# Local Residents Becoming Local Tourists: Value Co-Creation in Chinese Wetland Parks during the COVID-19 Pandemic

**Yaodong Zhu \*, Norzaidahwati Zaidin and Yibei Pu**

Hashim International Business School, Universiti Teknologi Malaysia, Johor Bahru 81310, Malaysia
\* Correspondence: zhuyaodong@graduate.utm.my

**Abstract:** Despite their ecological value, wetland parks can be expensive to preserve and maintain, so local governments endeavor to design financially sustainable models by exploiting the ecotourism value of wetland parks. This trend has been facilitated by telecommunication technologies that enable value co-creation. Unlike previous studies that primarily assume tourists to be outsiders far from home, this study addresses a unique situation: travel restrictions during the COVID-19 pandemic generated a unique ecotourism market for local residents. This study responds to the above issue by examining the factors responsible for local tourists' value co-creation intention. Specifically, we drew on the theory of planned behavior to develop an extended model to address the research objective. The hypothesized model was empirically tested using an online survey of 386 local tourists who traveled to a wetland park in the Liangping district of Chongqing, China. Our results suggest that social norms, destination awareness, experience expectations, and facilitating conditions could affect local tourists' attitudes, which further influences local customers' value co-creation intention. Moreover, social norms, destination awareness, and experience expectations could affect local tourists' perceived value of a wetland park, thus further influencing local customers' value co-creation intention. In doing so, we made interesting insights and implications for ecotourism at a local level. Drawing on our survey in a specific wetland park, we highlight how local tourists' attitude and perceived value positively affect their value co-creation intention and identify one more possible source of destination awareness: friends' sharing of destination information and experience through social media. Practically, we suggest local tourism to offset the maintenance costs of wetland parks during the COVID-19 pandemic. That requires leveraging social norms and understanding residents' expectations, in addition to improving infrastructure.

**Keywords:** wetland parks; local tourists; value co-creation intention; COVID-19 pandemic; theory of planned behavior

## 1. Introduction

A wetland refers to the areas of marsh, fen, peat land, and naturally or artificially formed water within a depth of six meters. These elements form a unique ecosystem that delivers ecological and economic values [1]. Economically, wetlands provide local residents with water resources, food, and income from ecotourism, i.e., a tourism activity that contributes to the local economy, educates tourists about local cultures, supports local development, and discourages mass construction [2]. As a form of ecotourism destination, wetland parks allow tourists to enjoy marine and freshwater recreation, adventure and cultural activities, camping, and hiking [3]. To fortify the ecological and economic value of wetland parks, governments have invested in the development and maintenance of wetland parks, together with the surrounding infrastructure and tourist attractions [4]. In the U.S., the Department of the Interior has financed, through the Fish and Wildlife Services, the conservation and restoration of 177,000 acres of wetland and bird upland habitats [5]. In China, the National Wetland Conservation Action Plan has listed 57 major wetlands (around 7 million hectares) for protection [6]. The Chinese central government's plan has

also motivated a range of stakeholders, including local governments (e.g., local forestry and grassland administration and local bureaus of tourism) and local investors. While it has developed into a rich experience in North America and Europe, ecotourism practices in developing countries have been underdeveloped [7]. In China, local government agencies are burdened with the high costs of preserving and maintaining infrastructure in existing wetland parks, with some undertaking to develop the local wetland parks to attract tourists and create jobs, thereby keeping these parks financially sustainable. According to ecotourism literature, tourists are motivated by the opportunity to observe and experience undisturbed and uncontaminated natural areas [8]. The advent of telecommunication technologies such as location-based guiding apps, tourism recommendation apps, and social media have collectively changed tourists' travel planning mode. In particular, tourists are actively involved in those technologies to explore, interact, and co-generate content along with ecotourism service providers and other tourists, thereby co-creating value [9]. Thus far, the tourism literature has investigated the value of co-creation activities between tourists and tourism service providers [10], tourists and local residents [11,12], and tourists themselves [13]. These studies assume that tourists are individuals who travel far away from home and spend time with local residents or other tourists in other areas [14]. However, this assumption was challenged during the COVID-19 pandemic, where lockdown policies kept local residents from traveling to other places; this resulted in a pent-up demand, which may be released locally.

Since 2020 and the beginning of the COVID-19 pandemic, governments worldwide have taken different degrees of lockdown and social distancing measures to prevent foreign tourists from entering [15]. These measures caused tremendous negative impacts on the global tourism, hospitality, and travel sectors. According to the 2021 report by the World Tourism Organization (UNWTO), the international tourist arrivals were 70–75% lower in 2021 than that in 2019, suggesting around a $1 trillion reduction in export revenues [16]. The reduced outward international travel stimulated the development of domestic tourism in China. According to a McKinsey & Company report, domestic air, rail, and sea travel has nearly returned to the pre-COVID-19 level, with 637 million pent-up Chinese customers traveling during the 7-day National Day holiday in October 2020 [17].

Indeed, the ongoing COVID-19 pandemic and the rising of social media platforms such as TikTok have provided precious opportunities for local government agencies to cater to the traveling need of local tourists. From wetland park tourism providers' perspective, they need not only interact with local tourists about the natural beauty but also provide superior tourism experiences. This is embedded in the interactions between local tourists and the various places, services, agencies, and cultures [18]. Despite their natural beauty and biodiversity, wetland parks are often located in nonurbanized areas with limited public transportation and entertainment (e.g., shopping and leisure services) [19]. Against this backdrop, wetland parks need proper marketing strategies to keep the local tourists interested and happy to visit. The literature has suggested tourism destination strategies, such as branding, cooperation, alliance, and partnership [20,21]. Moreover, local governments in China started to loosen the inter-provincial travel restrictions and helped local tourism destinations to improve exposure on social media platforms such as TikTok [22]. Social media platforms can help raise local tourists' awareness of local tourism destinations [23] by capturing visitors' experiences.

Despite the above efforts, studies on wetland parks have mostly focused on the strategies and efforts of ecotourism service providers [24,25], with limited attention to the factors responsible for local tourists' value co-recreation intention. This is problematic because a tourism service provider's perspective focuses more on profit generation than tourist value [26], and such a focus may result in a mismatch that eventually harms tourists' satisfaction, as well as their revisit intentions. Scholars have realized the importance of understanding how the various values delivered by ecotourism are actually perceived by tourists [27], with some relating tourists' value perceptions to ecotourism destination satisfaction and trust. Tourism studies have recognized tourists as passive recipients

of services but also active seekers of benefits and risks associated with specific tourism destinations [8]. In that case, tourist service providers should include tourists in the value co-creation process, explore the stimuli that could effectively lead to tourist satisfaction, and interact with tourists for closer relationships. Nevertheless, not much has been written about the factors contributing to tourists' value co-creation intentions for wetland parks. Among the sporadic wetland park studies that did include tourist behavior, the analysis was either descriptive (e.g., the important wetland park functions and landscapes perceived by tourists) [28] or general (e.g., tourists' compared evaluation of costs and emotional stimuli) [27]. This is problematic because, without a clear understanding of the specific factors that influence tourists' value co-creation intentions, wetland park organizers may be unable to provide the service portfolio to meet tourist needs [29,30].

Given the above research gap, this study aims to draw on the theory of planned behavior (TPB) to examine the antecedents of local tourists' value co-creation intentions during the COVID-19 pandemic in China, where international travel is still not available. The tourism literature has recognized TPB as instrumental for the exploration and verification of factors that predict tourists' behavioral intentions [31]. However, the original framework has not considered local tourists' expectations of the travel experience, in addition to the pre-conceived destination awareness prior to their arrival, although these factors could also influence tourists' attitudes and satisfaction [32]. Therefore, this study will develop and examine an extended conceptual model with adapted constructs in the TPB to reflect the unique research context.

## 2. Theoretical Background and Hypothesis Development

### 2.1. Value Co-Creation in Wetland Parks

Value-cocreation is based on the assumption that services involve the participation and interaction of service providers and service consumers [33] and that interactive stakeholders are equally important for the creation of value. Value co-creation is relevant to experience-specific contexts; it has been widely adopted by ecotourism researchers [8,34]. Interactive stakeholders include not only the wetland park organizations and local tourists but also location-based retailing app developers, government agencies, and service vendors around the wetland parks, such as local residents, hotels, hospitals, and other cultural heritage organizers. These stakeholders form a value co-creation platform where resources and services can be exchanged. By interacting with other stakeholders, local tourists would develop their own expectations about the experience in the specific area, thereby forming specific attitudes and perceived values that eventually lead to their value co-creation intention. Based on [35,36], value co-creation intention can be understood as a local tourist's intention to physically and virtually participate in ecotourism activities related to a specific wetland park destination.

### 2.2. Theory of Planned Behavior

According to Fishbein and Ajzen, one intends to behave in a specific way because he or she believes that this behavior can result in a specific result [37]. Such a process involves two key components: the person's attitude and subjective norms [38]. While the person's attitude shapes the belief that further changes his or her intention towards the behavior, subjective norms decide the normative beliefs, i.e., his or her beliefs that important referents encourage him or her to behave in particular manners towards the behavior [39,40]. Drawing on Fishbein and Ajzen, Ajzen added perceived behavioral control, i.e., the person's perception degree of difficulty in performing the behavior [41,42]; this is known as the theory of planned behavior (TPB), which has been widely adopted by researchers in explaining tourist behaviors [43,44]. However, researchers [45–48] increasingly argue that the explaining power of the TPB can be enhanced by incorporating new factors, decomposing existing factors, or modifying existing causal relationships based on the research contexts. As such, this study modifies the TPB model by decomposing the sources (e.g., destination awareness, experience expectation, and facilitating conditions) of local

tourists' attitudes, adapting subjective norms (i.e., social norms), and adding an additional variable (i.e., perceived value) to improve the explanatory power for predicting their value co-creation intention. Attitude can be understood as one's positive or negative evaluation of a specific object or behavior [49,50]. Attitude has been assumed to be significantly related to the individual's intention and behavior [51].

Therefore, the following can be hypothesized:

**Hypothesis 1.** *A tourist's attitude has a positive influence on his or her value co-creation intention.*

In addition to attitude, tourists' perceptions of the value of the experience may also affect their intentions. Value refers to the local tourist's assessment of the various benefits related to the travel experience based on his or her perception; it can be multivarious, including social, cognitive, emotional, hedonic, and utilitarian dimensions [52]. For instance, the COVID-19 travel restrictions may create a pent-up demand for recreation and entertainment [53]. Such postponed demand will be released once traveling within the region is lifted. Knowing that they can only travel locally, tourists may turn to local resorts that are less populated and with healthy recreational services for themselves and their families; moreover, they may perceive their traveling to local tourist sites as contributing to the local tourism industry and ecosystem protection, which they could share on social media. These perceived benefits could predict a local tourist's feelings on how the experience could deliver the various values to meet their expectations, thereby predicting his or her behavioral intentions [54]. Therefore, the following can be hypothesized:

**Hypothesis 2.** *A tourist's perceived value has a positive influence on his or her visit intention.*

The following part of this section investigates the antecedents of attitude and perceived value based on the TPB model.

*2.3. Social Norms*

Studies on individuals' intention to reduce environmental impact have adopted social norms [55,56]. Social norms can be descriptive and injunctive [57], with the former indicating the mostly accepted behavior learned by observing how others perform it, while the latter indicating the degree to which behavior will receive the moral approval of referees [58,59]. Both descriptive norms and injunctive norms can influence local tourists' behaviors. When their cities have zero COVID-19 cases, local tourists may decide not to travel to other countries or cities and instead visit local tourist sites because it suggests that others have been making the same efforts to prevent the cross-regional spread of the virus (descriptive norms). In contrast, when their cities have zero COVID-19 cases, local tourists may decide not to visit other countries or cities and instead visit local tourist sites because it suggests that visiting across regions would be criticized and even punished by local residents. Both types of social norms could shape local tourists' attitudes toward a tourist destination. Social norms related to tourism during the COVID-19 pandemic may influence local tourists' attitudes towards local tourist destinations. Now that destinations in other cities or countries become unavailable, local tourists may develop preferable attitudes towards local tourist destinations to demonstrate their compliance with the aforementioned social norms during the pandemic. Moreover, local residents may listen to their friends and families regarding the benefits such as discounts, better services (due to reduced tourist population), the fun of spending time with important ones, and even contribute to local tourism.

Therefore, the following can be predicted:

**Hypothesis 3.** *Social norms have a positive influence on a tourist's attitude towards a local wetland park.*

**Hypothesis 4.** *Social norms have a positive influence on a tourist's perceived value toward a local wetland park.*

### 2.4. Destination Awareness

The extensive advertisements from wetland park organizers and local media could promote tourists' awareness of a destination, i.e., a potential tourist's ability to recognize and remember a specific destination as a tourism choice [60]. The literature has recognized the importance of destination awareness in communicating information with potential tourists for the purpose of transaction (e.g., visit) [61]. The degree of a tourist's awareness of a specific tourist destination reflects the destination's strength in the potential tourist's mind, so it is essential for a tourist destination [62]. As COVID-19 struck each tourist destination, local governments and destination organizers took active means to promote wetland parks through exposure on social media and by adopting former tourists' generated content [63]. The visual and audio appeals regarding wetland destinations reflect the quality of a specific tourist destination, and they are associated with potential tourists' perceptions of the benefits and expectations of a specific destination [64]. However, with travel restrictions in place, the destination organizers' attention has turned to local tourists, reminding them of the various benefits that are available in the local tourist destinations and encouraging them to visit local tourist destinations, which used to bear the expectation of over-crowdedness and tourists from other regions.

**Hypothesis 5.** *Destination awareness has a positive influence on a tourist's attitude towards a local wetland park.*

**Hypothesis 6.** *Destination awareness has a positive influence on a tourist's perceived value of a local wetland park.*

### 2.5. Experience Expectations

Expectation refers to an individual's belief that enacting a specific behavior will lead to better outcomes [65,66]. The anticipated satisfaction and the likelihood that an act is positively associated with a specific outcome collectively influence one's intention to behave in a specific manner [67]. Tourists' behaviors can be regarded as the results of tourists' engagement with the expected outcomes or rewards [68]. Some scholars suggest that a complete tourist experience involves several phases, from pre-visit expectations to after-visit memories [69]. We define tourist expectations as the results of tourists and tourism destinations before the actual visit. This can be achieved through tourists' virtual experiences, such as pushed recommendations from location-based consumption apps and leaflets in local communities. These sources exert various advertising effects that influence local tourists' expected experiences in a specific location (by improving their familiarity with the destination) [70]. With such familiarity and expectation, local tourists are more likely to form favorable attitudes and perceive the value of visiting local wetland parks [71]. Therefore, the following can be predicted:

**Hypothesis 7.** *Experience expectation has a positive influence on a tourist's attitude towards a local wetland park.*

**Hypothesis 8.** *Experience expectation has a positive influence on a tourist's perceived value of a local wetland park.*

### 2.6. Facilitating Conditions

A consumer's perceived behavioral control in TPB can be reflected in his or her judgment of the ease of enacting a certain action [72]. In local tourism, perceived behavioral control depends on facilitating conditions, such as the cost, effort, and time required for the travel experience. Facilitating conditions involve individuals' perception of their autonomy during their consumption behavior [73,74]. The availability of critical resources could provide local tourists with a high degree of autonomy, thus lowering obstacles to enacting a behavior. In other words, tourists' attitudes towards visiting a wetland park could be

affected by objective conditions such as resources and policies. In the context of this study, facilitating conditions involve the cross-regional travel restrictions which keep tourists from external areas away from the local wetland park, thereby reducing the crowdedness for local tourists, the convenience of public transport linking urban areas to the wetland parks, as well as the various discounts that local organizers provide to recover the local tourism, as well as the location-based consumption apps that allow local tourists to choose to visit during low seasons. Therefore, the following can be predicted:

**Hypothesis 9.** *Facilitating conditions have a positive influence on a tourist's attitude towards a local wetland park.*

**Hypothesis 10.** *Facilitating conditions have a positive influence on a tourist's perceived value of a local wetland park.*

The above hypotheses form the conceptual model of this study (see Figure 1).

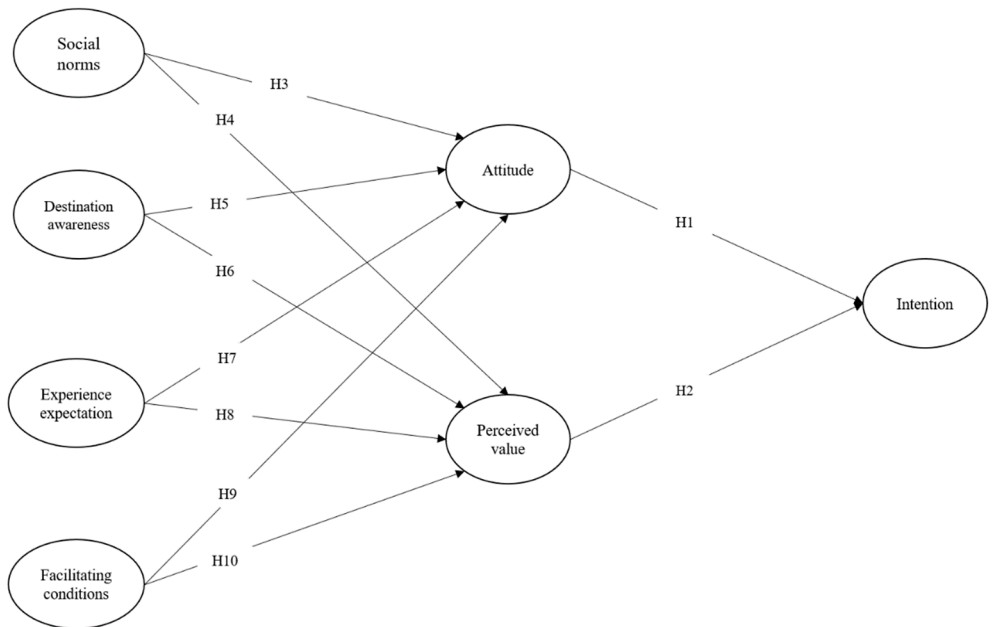

**Figure 1.** Conceptual model.

## 3. Methods

### 3.1. Sampling

To test the research hypotheses, we hired local travel agencies and hotel managers who conducted an online survey of customers of three locally known hotels: Days Hotel Days Hotel & Suites Liangping (DHS), Chongqing Yishanjun Grand Hotel (YGH), and Pingye Yuanlin Hotel (PYH) in Liangping district, Chongqing China. These hotels were close (DHS 2.1 km, YGH 5.1 km, and PYH 5.7 km) to Shuanggui Lake National Wetland Park (SWP), which covers an area of 349.97 hectares, with the second largest lake in the city. It is home to 207 vertebrate species (including endangered bird species such as aythya baeri/Baer's pochard and aythya farina/common pochard). Guests of these hotels could travel to SWP by bicycle, taxi, bus, or shared car in 20 min.

SWP is located in the north of Chongqing municipality, which has 32.8 million residents, among whom 17 million live in the urban area. These urban residents form a tremendous tourism market that local tourist sites such as SWP seek to attract. While SWP is over 200 km from the city center, it is within one hour of access to city residents by high-speed train.

To address our research context, we first asked each respondent to confirm their origins and actual visits to SWP, with non-local residents and non-visitors excluded from the survey. The survey was conducted from July to August 2021, when local residents were encouraged to travel within the city. With the help of local travel agencies, we obtained consent from 473 respondents to conduct the survey. After the survey, we scrutinized the data for missing values and outliers, and this effort led to the deletion of 67 responses, leaving 386 usable surveys. Thus, the final sample of this study comprised 386 local tourists. Out of the 386 respondents, 62.2% were male and 37.8% female. Regarding the age group, 30.3% were 18–30 years old, 34.7% were 31–40 years old, 28.8% were 41–50 years old and 6.2% were aged 50 years and above. Regarding education level, 21% had a college diploma or below, 28.5% had a bachelor's degree, 42.7% had a master's degree, and 7.8% had a doctorate degree. In terms of monthly income, 25.1% earned 3000 yuan and below; 21.2% earned 3001–4500 yuan; 22.5% earned 4501–6000 yuan; and 31.1% earned 6001 yuan and above. Regarding locations, 58.3% were from the same district (i.e., Liangping district), and 41.7% were from other districts of Chongqing. Table 1 reported the demographic data of the respondents. Table 1 provides a summary of respondents' demographic profiles.

**Table 1.** Profile of the respondents.

| | Variable | Frequencies | Percentage |
|---|---|---|---|
| Gender | Male | 240 | 62.2% |
| | Female | 146 | 37.8% |
| Education | College or below | 81 | 21.0% |
| | Bachelor degree | 110 | 28.5% |
| | Master degree | 165 | 42.7% |
| | Doctorate degree | 30 | 7.8% |
| Age | 18–30 years old | 117 | 30.3% |
| | 31–40 years old | 134 | 34.7% |
| | 41–50 years old | 111 | 28.8% |
| | 50 years old and above | 24 | 6.2% |
| Income | 3000 yuan and below | 97 | 25.1% |
| | 3001–4500 yuan | 82 | 21.2% |
| | 4501–6000 yuan | 87 | 22.5% |
| | 6001 yuan and above | 120 | 31.1% |
| Location | Liangping district | 225 | 58.3% |
| | Other districts | 161 | 41.7% |

*3.2. Measures*

The questionnaire for this study included seven constructs: social norms, destination awareness, experience expectation, facilitating conditions, attitude, perceived value, and intention to co-create value. All of those measures were adapted from existing measures validated by previous research. All of the items (see Appendix A for all the items) were measured on 5-point Likert scales (1: strongly disagree; 5: strongly agree).

- Social norms

The scale for social norms (SN) was adapted from Izquierdo-Yusta, Martínez–Ruiz [75]. The five items in this scale were revised and adjusted to meet the research scope of this study. An example item includes, 'The people who are important to me (family and friends) think I should visit here.' The Cronbach's alpha of this scale was 0.919.

- Destination awareness

Destination awareness (DA) was adapted from the 6-item measurement scale in [76]. Example items of this scale include 'I can tell the unique specialty of this wetland park among other competing wetland parks' and 'Some characteristics of this wetland park come to my mind quickly'. The Cronbach's alpha of this scale was 0.929.

- Experience expectation

The scale for experience expectation (EE) was adapted from the 10-item measurement scale in Sheng and Chen [77]. 'During the trip, I expect to be relaxed, such as taking my time walking or visiting friendly environments' and 'During the trip, I expect to find some interesting contrasts and changes, such as changes from the city life, closeness to nature, and even some unique activities' were examples of this scale. The Cronbach's alpha of this scale was 0.949.

- Facilitating conditions

The scale for facilitating conditions (FC) was adapted from the 3-item scale in Yang and Forney [78]. 'Given the transportation, discount information, and knowledge I have about this park, it would be easy for me to travel here' and 'I have the knowledge necessary for a trip here' were examples of this scale. The Cronbach's alpha of this scale was 0.813.

- Personal attitude

The scale for personal attitude (PA) was adapted from the 3-item scale in Prados-Peña and Del Barrio-García [79]. 'It strikes me as a good wetland park' and 'I like this park' were examples of this scale. The Cronbach's alpha of this scale was 0.852.

- Perceived value

The scale for perceived value (PV) was adapted from Dodds, Monroe [80]. 'This park is a very good value for the money' and 'The consumptions I made around the park are very economical' were examples of this scale. The Cronbach's alpha of this scale was 0.934.

- Value cocreation intention

The scale for value cocreation intention (VCI) was adapted from Rather, Hollebeek [30]. 'I have used my experience from previous visits in order to arrange this visit' and 'I have been actively involved in the destination co-creation experience' were examples of this scale. The Cronbach's alpha of this scale was 0.870.

- Control Variables

In selecting variables to include as controls, we focused on those variables that could potentially be viewed as alternative explanations for personal attitude, perceived value, or value co-creation intention. Therefore, we followed [81] to control the gender, education, age, income, and location of each respondent.

## 4. Results

### 4.1. Reliability and Validity

We adopted confirmatory factor analysis (CFA) to validate the measurement model. This was achieved by assessing the reliability, convergent validity, and discriminant validity of the constructs. According to our analysis, the fit statistics of the measurement model had good fit indices ($\chi^2$ = 975.942, DF = 573, CFI = 0.962, TLI = 0.959, RMR = 0.048, and RMSEA = 0.043). Cronbach's alpha for each latent construct ranged from 0.813 to 0.949, exceeding the recommended level of 0.70.

Convergent validity was ensured with composite reliability (C.R.) above 0.8 and AVEs over 0.5 [82]. According to Table 2, all C.R. values are higher than the suggested 0.80, and all AVE values are higher than the suggested 0.50, thereby indicating a good convergent validity of the measurement model. Discriminant validity among constructs was confirmed by comparing the squared root of the AVE for each construct with the correlations between that construct and all other constructs. In Table 3, the squared roots of the AVEs for the constructs that were greater than the correlations between a given construct and others satisfied the discriminant validity. Thus, the discriminant validity among constructs was achieved.

**Table 2.** Validity and reliability.

| | Items | STD. Estimate | CR | AVE | Cronbach's α |
|---|---|---|---|---|---|
| Social norms | SN1 | 0.717 | 0.921 | 0.701 | 0.919 |
| | SN2 | 0.884 | | | |
| | SN3 | 0.781 | | | |
| | SN4 | 0.888 | | | |
| | SN5 | 0.899 | | | |
| Destination awareness | DA1 | 0.786 | 0.929 | 0.687 | 0.929 |
| | DA2 | 0.901 | | | |
| | DA3 | 0.844 | | | |
| | DA4 | 0.815 | | | |
| | DA5 | 0.824 | | | |
| | DA6 | 0.797 | | | |
| Experience expectation | EE1 | 0.753 | 0.950 | 0.657 | 0.949 |
| | EE2 | 0.813 | | | |
| | EE3 | 0.846 | | | |
| | EE4 | 0.797 | | | |
| | EE5 | 0.925 | | | |
| | EE6 | 0.739 | | | |
| | EE7 | 0.787 | | | |
| | EE8 | 0.872 | | | |
| | EE9 | 0.766 | | | |
| | EE10 | 0.787 | | | |
| Facilitating conditions | FC1 | 0.777 | 0.814 | 0.593 | 0.813 |
| | FC2 | 0.771 | | | |
| | FC3 | 0.762 | | | |
| Attitude | PA1 | 0.873 | 0.851 | 0.656 | 0.852 |
| | PA2 | 0.813 | | | |
| | PA3 | 0.738 | | | |
| Perceived value | PV1 | 0.855 | 0.940 | 0.759 | 0.934 |
| | PV2 | 0.817 | | | |
| | PV3 | 0.904 | | | |
| | PV4 | 0.917 | | | |
| | PV5 | 0.859 | | | |
| Value cocreation intention | VCI1 | 0.866 | 0.872 | 0.632 | 0.870 |
| | VCI2 | 0.839 | | | |
| | VCI3 | 0.704 | | | |
| | VCI4 | 0.760 | | | |

Note: $n$ = 386.

**Table 3.** Correlation and discriminant validity analysis.

| | Mean | SD | 1 | 2 | 3 | 4 | 5 | 6 | 7 | 8 | 9 | 10 | 11 | 12 |
|---|---|---|---|---|---|---|---|---|---|---|---|---|---|---|
| 1. Gender | 1.378 | 0.486 | - | | | | | | | | | | | |
| 2. Education | 2.373 | 0.901 | 0.092 | - | | | | | | | | | | |
| 3. Age | 2.109 | 0.911 | −0.046 | −0.148 ** | - | | | | | | | | | |
| 4. Income | 2.596 | 1.170 | 0.073 | −0.024 | −0.068 | - | | | | | | | | |
| 5. Location | 1.417 | 0.494 | 0.044 | 0.152** | −0.251 ** | −0.009 | - | | | | | | | |
| 6. SN | 3.600 | 0.864 | 0.080 | 0.072 | −0.001 | 0.002 | 0.016 | 0.837 | | | | | | |
| 7. DA | 3.467 | 1.046 | 0.065 | 0.070 | 0.042 | 0.004 | −0.066 | 0.546 ** | 0.829 | | | | | |
| 8. EE | 3.744 | 1.005 | 0.002 | 0.029 | −0.004 | 0.023 | −0.007 | 0.495 ** | 0.573 ** | 0.810 | | | | |
| 9. FC | 3.725 | 1.039 | 0.078 | 0.088 | −0.015 | −0.020 | 0.014 | 0.451 ** | 0.460 ** | 0.462 ** | 0.770 | | | |
| 10. PA | 3.852 | 0.856 | 0.085 | 0.000 | −0.025 | −0.005 | −0.067 | 0.402 ** | 0.402 ** | 0.431 ** | 0.380 ** | 0.810 | | |
| 11. PV | 3.752 | 0.938 | 0.063 | −0.003 | 0.093 | −0.092 | −0.042 | 0.532 ** | 0.426 ** | 0.401 ** | 0.331 ** | 0.355 ** | 0.871 | |
| 12. VCI | 3.709 | 0.916 | −0.014 | −0.022 | −0.033 | 0.065 | −0.036 | 0.317 ** | 0.335 ** | 0.377 ** | 0.376 ** | 0.354 ** | 0.277 ** | 0.795 |

Notes: $n$ = 386; **, $p < 0.01$; SN, social norms; DA, destination awareness; EE, experience expectancy; FC, facilitating conditions; PA, personal attitude; PV, perceived values; VCI, value co-creation intention.

### 4.2. Common Method Bias

To check the problem of common method bias, we conducted Harman's single-factor test. The analysis returned seven factors with eigenvalues greater than 1, with the first factor explaining less than 40% of the variance (39.678% of 74.474%). Thus, the findings provided no serious indications of common method variance.

### 4.3. Hypothesis Testing

The results of the standardized coefficients for each hypothesized path are provided in Table 4.

**Table 4.** Regression analysis.

| | P.A. | | PV | | VCI | |
|---|---|---|---|---|---|---|
| | Model 1 | Model 2 | Model 3 | Model 4 | Model 5 | Model 6 |
| Gender | 0.088 | 0.058 | 0.074 | 0.035 | −0.017 | −0.057 |
| Education | −0.003 | −0.042 | 0.004 | −0.039 | −0.019 | −0.019 |
| Age | −0.043 | −0.049 | 0.084 | 0.074 | −0.043 | −0.046 |
| Income | −0.015 | −0.017 | −0.092 | −0.094 * | 0.062 | 0.084 |
| Location | −0.081 | −0.071 | −0.025 | −0.018 | −0.043 | −0.015 |
| SN | | 0.163 ** | | 0.386 *** | | |
| DA | | 0.117 * | | 0.120 * | | |
| EE | | 0.214 *** | | 0.123 * | | |
| FC | | 0.153 ** | | 0.044 | | |
| PA | | | | | | 0.290 *** |
| PV | | | | | | 0.190 *** |
| $R^2$ | 0.014 | 0.274 | 0.022 | 0.340 | 0.008 | 0.164 |
| F | 1.08 | 15.741 *** | 1.706 | 21.475 *** | 0.587 | 10.574 *** |

Notes: $n$ = 386; *, $p < 0.05$; **, $p < 0.01$; ***, $p < 0.001$; SN, social norms; DA, destination awareness; EE, experience expectancy; FC, facilitating conditions; PA, personal attitude; PV, perceived values; VCI, value co-creation intention.

Based on our regression analysis (as set in Table 4), most of the hypotheses were supported, except for Hypothesis 7. First, local tourists' favorable attitude has a significant positive effect on their intentions to co-create value (β = 0.290, $p < 0.001$); Hypothesis 1 was supported. Likewise, the perceived value was also found to exert a significantly positive effect on their intentions to co-create value (β = 0.190, $p < 0.001$), so Hypothesis 2 was supported.

Regarding the influence of social norms (SNs), SNs were found to positively influence tourists' attitudes toward the local wetland park (β = 0.163, $p < 0.01$), so Hypothesis 3 was supported. This result suggests that the external sources (e.g., family or friends' preferences and social consensus regarding travel during COVID-19) collectively provide the information and expectations for local tourists to adjust their attitudes towards local wetland parks. SNs were also found to positively influence local tourists' perceived value related to local wetland parks (β = 0.386, $p < 0.001$), so Hypothesis 4 was supported. These

results reflect how external comments and opinions on a specific wetland park could affect local tourists' evaluation of the benefits of travel experience to local wetland parks. While individuals may form different perceived values towards the same products or services [83], our results demonstrate how such values can be influenced by the important individuals in their lives through positive stimuli (e.g., family members and friends) and negative reinforcement (e.g., local travel restriction policies).

Regarding the impact of destination awareness (DA), DA was found to exert a significantly positive effect on tourists' attitudes toward the local wetland park ($\beta = 0.117$, $p < 0.05$), so Hypothesis 5 was supported. As tourists search for information about local tourism sites, their awareness of a specific site becomes the essential prerequisite to their attitudinal preference. In the literature [79,82], destination awareness reflects how tourists think they know about a destination. However, our study suggests that in the digital media age, when tourism sites compete against each other in depicting scenic views and comfortable living experiences, tourists may experience an overload of information and may not necessarily form preferential attitudes towards a specific site. This study captures a unique context where tourism sites in other cities become unavailable due to travel restrictions; in that case, the destination awareness in this study is limited by the immediate accessibility for pent-up tourist demand. Moreover, DA was found to positively influence tourists' perceived value related to local wetland parks ($\beta = 0.120$, $p < 0.05$), so Hypothesis 6 was supported. This concurs with the consumer behavior studies on the impact of brand awareness and perceived value [83]. In the context of this study, the association could be explained by the updated audio-visual information about the local wetland park, the improved public transportation, as well as the promotion information provided by park organizers.

Regarding the impact of experience expectation (EE), EE was found to exert a significantly positive impact on tourists' attitudes toward the local wetland park ($\beta = 0.214$, $p < 0.001$), so Hypothesis 7 was supported. This result concurs with [69], who considered the multiple stages of the tourist experience, starting with expectations. EE has a significant positive effect on tourists' perceived value related to local wetland parks ($\beta = 0.123$, $p < 0.05$), so Hypothesis 8 was supported. In this study, tourists' experience expectations are related to their pent-up demand after the COVID-19 lockdown and social distancing, which influences their tourism preferences and cognitions. For instance, tourists wary of the COVID-19 virus may picture themselves taking recreational activities in less crowded areas and select tourist sites that meet their expectations (i.e., forming preferential attitudes and perceived value concerning a specific tourism site).

Regarding the impact of facilitating conditions (FCs), FCs were found to exert a significantly positive effect on tourists' attitudes toward the local wetland park ($\beta = 0.153$, $p < 0.01$), so Hypothesis 9 was supported. This result concurs with [74] regarding how facilitating conditions could enhance consumers' attitudes toward a specific product or service by enabling autonomy and resources. Surprisingly, our empirical results did not find a significant relationship between FC and tourists' perceived value related to local wetland parks ($\beta = 0.044$, $p > 0.05$), so Hypothesis 10 was not supported. This surprising finding suggests that facilitating conditions alone are not considered to generate the various dimensions of value to tourists.

## 5. Discussion

The local government, in our case, has invested generously in the wetland park, yet later found it challenging to keep these parks financially sustainable. As such, the local tourism and forestry management agencies added recreational and educational functions to attract tourists and create jobs. This seems especially important during the COVID-19 pandemic in this area, where traditional tourism options (e.g., cross-country and cross-regional travels) were still unavailable, thereby generating negative impacts on the local tourism and hospitality industries. Currently, China still adopts the 'dynamic static management of COVID-19 risks', meaning that if any COVID-19 case is found, local residents will be

instructed not to leave their local residences. This leads tourists to delay their consumption of ordinary services such as long-distance travel because the tourism market cannot provide for their demands for various tourism experiences, i.e., the pent-up demand. While tourism organizers may consider this as a great market opportunity to cater to local tourists, they very often take a focal perspective, i.e., investigating the strategies and benefits of tourism service providers. Given this knowledge gap, this study investigates the antecedents of local tourists' value co-creation intention during the COVID-19 pandemic in a wetland park in China. Specifically, we explored the relationships between tourism social norms, destination awareness, experience expectation, facilitating conditions, local tourists' attitudes, perceived value, and value co-creation intention. Given this unique context, we developed our modeled associations and examined the interactions among these variables to generate new insight.

### 5.1. Theoretical Implications

This study makes several theoretical contributions. First, our study suggests that, to keep the wetland park projects financially sustainable, organizers and policymakers should integrate ecotourists into the value co-creation process. Specifically, this study highlights local tourists' perceived autonomy in their value co-creation intention. Second, drawing on the descriptive and injunctive nature of norms, we adopted social norms [55,56] to demonstrate how observations of other people's behavior or habits form an additional source of norms that can influence tourists' attitudes and perceived value in our selected case. Third, many ecotourism destinations (e.g., other wetland parks in Chongqing) compete for local tourists when external tourists are no longer available during the COVID-19 pandemic. In addition to social media and interactive location-based consumption apps, this study suggests one more possible source of destination awareness: friends' sharing of destination information and experience through social media further promotes the destination awareness of local tourists. Suggestions and recommendations from important individuals (e.g., friends and families) may positively influence local tourists' attitudes and perceived value, which further promotes their value co-creation intentions. Likewise, friends' sharing may also facilitate experience expectation as an important antecedent of tourists' attitudes and perceived value related to local wetland parks before traveling.

### 5.2. Practical Implications

Our research findings have some practical implications for ecotourism service providers and policymakers and location-based recommendation platform developers in China during the COVID-19 pandemic. First, given the tremendous costs of maintaining wetland parks, service providers and policymakers could develop ecotourism activities that generate revenue to offset their daily operations. In particular, the current 'dynamic static management of COVID-19 risks' in China has unwittingly developed a tremendous market for domestic and especially local tourism. Wetland parks that are located in the suburbs of large metropolitan areas could consider raising local residents' consciousness through sharing among friends and families.

Second, service providers should understand and leverage the social norms that could influence local tourists' attitudes toward and perceived value of wetland parks. Our study found social norms coming from reunions with important people (e.g., families and friends) as well as a healthy lifestyle during the COVID-19 pandemic. Third, while the local governments in our study have invested generously in improving local infrastructure and services, policymakers and wetland park service providers should understand that such efforts may not necessarily improve local tourists' perceived value related to the destinations. Indeed, as ecotourism sites compete for tourists on social media, the above-mentioned facilitating conditions seem to be fundamental, if not determining, to users' value co-creation intentions. In other words, the facilities and transportation to a specific wetland park are the basic requirements for local governments to seize the local tourism opportunities mentioned in this study.

*5.3. Limitations and Future Research Directions*

Despite the above conclusion and implications, this study is subject to some limitations. First, we only investigated local tourists' value co-creation intentions rather than their specific value co-creation practices. Although intentions are determinant of individual behaviors, the relationship might be affected by contingent factors, which future research could include to provide a fuller picture of the process. Second, due to the purpose of this study being to investigate local tourists' value co-creation intentions, non-local tourists were excluded from the study. Future studies could compare local and non-local tourists by examining the explaining power of our proposed conceptual model, thereby developing a more generalizable conclusion. Finally, we only used cross-sectional data to examine antecedents of local tourists' value co-creation intentions; future studies could consider qualitative interviews, observations, and text-mining to develop a richer and more in-depth understanding of the phenomenon covered in this study.

## 6. Conclusions

Drawing on the theory of planned behavior, this study suggests that local residents' attitudes and perceived value positively influence their value co-creation intention towards local wetland parks during the COVID-19 pandemic, when cross-city travel was inconvenient and international travel discouraged. We also found that social norms, destination awareness, experience expectation, and facilitating conditions are important antecedents of local residents' attitudes, as each of them is positively associated with it.

**Author Contributions:** Conceptualization, Y.Z. (Yaodong Zhu) and N.Z.; formal analysis, Y.Z. (Yibei Pu). and Y.Z. (Yaodong Zhu); investigation, Y.Z. (Yibei Pu). and Y.Z. (Yaodong Zhu); writing—original draft preparation, Y.Z. (Yaodong Zhu); writing—review and editing, Y.Z. (Yaodong Zhu); supervision, N.Z. All authors have read and agreed to the published version of the manuscript.

**Funding:** This research received no specific grant from any funding agency in the public, commercial, or not-for-profit sectors.

**Institutional Review Board Statement:** The study was conducted in accordance with the Declaration of Helsinki, and approved by the Research Ethics Committee (REC) Non-Clinical of Universiti Teknologi Malaysia (on 1 May 2021).

**Informed Consent Statement:** Informed consent was obtained from all subjects involved in the study.

**Data Availability Statement:** The data presented in this study are available on request from the corresponding author. The data are not publicly available due to privacy concerns.

**Conflicts of Interest:** No conflict of interest applied to this study.

## Appendix A. Measures

| Variables | Items |
| --- | --- |
| SN1 | The people who are important to me (family and friends) think I should visit this wetland park. |
| SN2 | The people who influence my behavior as opinion leaders (celebrities and experts) have talked about their experience in this wetland park. |
| SN3 | People whose opinions I value suggest that I visit this wetland park. |
| SN4 | The people that are important to me (family and friends) think that I should visit this wetland park as soon as possible. |
| SN5 | The people whose opinions I value would applaud my choice to visit this wetland park as soon as possible. |
| DA1 | I had known what this wetland park looked like before I came here. |
| DA2 | I can recognize this wetland park among other competing tourist sites. |
| DA3 | I was aware of this wetland park before I came. |
| DA4 | Before my visit, some characteristics of this wetland park had already come to my mind quickly. |
| DA5 | Before my visit, I could quickly recall the landform of this wetland park. |
| DA6 | Before my visit, I had little difficulty imagining this wetland park in my mind. |

| Variables | Items |
|-----------|-------|
| SN1 | The people who are important to me (family and friends) think I should visit this wetland park. |
| EE1 | Before my visit, I expected a relaxing experience in this wetland park, such as taking a walk or visiting friendly environments. |
| EE2 | Before my visit, I expected to find something interesting in this wetland park, such as rare birds and special activities. |
| EE3 | Before my visit, I expected to experience healthy and educational activities in this wetland park, such as natural environment preservation courses. |
| EE4 | Before my visit, I expected to see architecture and food with local cultural characteristics and varied images. |
| EE5 | Before my visit, I expected to be able to approach the core features of the wetland park, such as taking pictures of the rare animal and plant species. |
| EE6 | Before my visit, I expected to be identified, such as traveling with friends and families with similar interests. |
| EE7 | Before my visit, I expected to be close to the rare animal and plant species of my memories. |
| EE8 | Before my visit, I expected to experience a closeness to the elements of the wetland park. |
| EE9 | Before my visit, I expected to have a fantasy experience that has an area that resembles the wilderness I had watched in T.V. programs. |
| EE10 | Before my visit, I expected to enjoy the fulfillment of hopes or visions, such as visiting an unpolluted environment. |
| FC1 | I have a location-based consumption app to access all the services around this area. |
| FC2 | Given the transport, cost, and knowledge it takes to visit this wetland park, it would be easy for me to enjoy the trip. |
| FC3 | I know this area well enough to enjoy myself. |
| PA1 | This wetland park strikes me as a good park. |
| PA2 | I think this is a pleasant wetland park. |
| PA3 | I like this wetland park. |
| PV1 | This wetland park is very good value for the money. |
| PV2 | The consumption I made around the park was very economical. |
| PV3 | The park is considered to be a good place to visit. |
| PV4 | The costs associated with this wetland park are very acceptable. |
| PV5 | The costs of this park appear to be a bargain. |
| VCI1 | I will use my experience from this visit in order to arrange future visits. |
| VCI2 | I will be actively involved in the co-creation experience related to this wetland park. |
| VCI3 | I intend to discuss this wetland park's co-creation experience with others. |
| VCI4 | I have spent a considerable amount of time arranging this visit. |

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
