# Peer review of "Local Residents Becoming Local Tourists: Value Co-Creation in Chinese Wetland Parks during the COVID-19 Pandemic"

_sustainability, doi:10.3390/su141912577_

Round 1

Reviewer 1 Report

Dear Editors/Authors,

Presented manuscript is dealing with touristic value of the protected wetland areas, examining the factors responsible for local tourists' value co-creation intention. The topic  is interesting in the frame of COVID19-related changes in touristic attitudes towards attractions and expectations, that is important for the proper management and financial sustainability of the designated areas.

Major issues are related to methodology and how Authors relate the obtained data to actual visit (previous or current) of the questioned participants together with not clear presentation of their initial goal/intent to visit the area in first place. Several factors as the park proximity and accessibility that could be important for the intention are not accounted as hotels involved were described as five-star establishments within the city limits. Additionally measurement of impact of social-media remains unclear. Its is understandable that remote surveys have some limitations, however, it would be good to elaborate more in this direction.

Find comments and suggestions in the attached file.

Best regards,

TI

Author Response

Dear professor,

Please accept our sincere thanks for your constructive comments and suggestions for our manuscript. We revised the manuscript according to your comments, specifically:

  • On page 2, we corrected ‘local bureau’ into ‘local bureaus’
  • We provided the names and distance of the three hotels to the wetland park, together with their locations on Google Maps. Please find the details on pp.12-14.
  • We included the endangered birds’ names (aythya baeri/ Baer's pochard & aythya farina/common pochard) on page 12.
  • ‘Is wetland area accesible for city tour participants? Do Authors veryfy there is actual visit of the wetland park?'
  • We clarified this issue on page 14. ‘To address our research context, we first asked each respondent to confirm their origins and actual visits to SWP, with non-local residents and non-visitors excluded from the survey.’
  • In our Chinese version survey, we referred to ‘Associate degree’ as 2/3-year colleges. So we changed the English version into ‘college or below’. (page 15)
  • Full names of variables were added in Table 2 (page 18)
  • ‘Our findings about tourists’ positive responses to social media-enabled tourism to local wetland parks suggest that the process of value co-creation requires the participation of all stakeholders related to the local wetland parks.’ How do Authors measure the positive response to social media posts/info of the participants?'
  • Thanks for the comments. We have now removed such an argument, as we considered it an overgeneralization.
  • ‘Before my visit, I could quickly recall the shape of this wetland park.’ How shape of the park is important for its attractiveness or the intention to visit/co-create?' We double checked the translation, and corrected ‘shape’ into ‘landform’, as that’s what the Chinese survey actually meant.

Please find the revised manuscript in the attachment.

Kind regards,

Reviewer 2 Report

Re: Local Residents Becoming Local Tourists: Value Co-creation Intentions in Wetland Parks During the COVID-19 Pandemic in China: An Extension of the Theory of Planned Behavior

After carefully reviewing this manuscript, I think this is an interesting paper and this topic about value co-creation intentions in wetland parks is worthy of publication. I only have some small suggestions to help improve the quality of this paper. My suggestions are as follows:

1)    Paper structure

The section of literature review is very long so then the method section comes too late. I would suggest the authors further refine the ideas from literature, improving the readability of the paper.

2)    Discussion and conclusion

The authors combine discussion and conclusion into one section. I would suggest the author separate these two parts. For now, the section of results is not presented in detail enough, so adding a section of discussion could help enhance the findings and theoretical discourse of this research. While then, the conclusion section is only to present what had been approved in this paper, can be much clearer.  

Thanks editors and authors for letting me review this interesting paper, and I wish you good luck.

Author Response

Dear professor,

Please accept our sincere thanks for your constructive comments and suggestions for our manuscript. We revised the manuscript according to your comments, specifically:

1) Paper structure: We modified the paper structure by refining the literature review as well as the order of hypotheses for better readability (pp. 5-11).

2) Discussion and conclusion: we followed your suggestion to separate the Discussion section (pp. 22-25) from the Conclusion section (pp. 25-26) to improve clarity.

Please find the revised manuscript in the attachment.

Kind regards,

Reviewer 3 Report

Thank you for the opportunity to review this manuscript.  I offer the following comments

Subject matter

This is a v interesting paper and presents new insights on tourists' responses to Covid.  The study of local residents becoming local tourists captures a valuable insight into changed tourist behaviour while in the pandemic, and may lead to a re-think even of the meaning of 'tourist' (existing definitions envisage trips further afield than one's own city).

Literature and background

The manuscript has a very broad foundation in the literature, but I think you need to re-consider the appropriateness of some items.  For example:

Ref 2 is your source for detail on 'ecotourism': I think a more mainstream article with a stronger tourism focus, and including a suitable definition, such as Cobbinah (2015) 'Contextualising the meaning of ecotourism' would be more appropriate

Ref 4 relates to ecosystem services - maybe a more applicable source on this point would be say Keenan et al (2019) 'Ecosystem services in environmental policy', or Maes et al (2019) 'Ecosystem services are inclusive and deliver multiple values'.

P 5 Line 21 'We define tourist expectations as the results of ...': no source is provided for this statement, and I suggest this may be a contentious point which readers would need to be convinced of

Ref 67, used at the foot of P 5 in regard to local tourists attitudes: this ref is written in the context of climate change and tourism.  While relevant I would seek down a more general source defining tourists attitudes.

Methodology

The methodology and analysis is well chosen and implemented. The case study approach is used well (but see comments below).

However, a number of issues concern me:

The sample size is v small and v specific, and I believe these constraints reduce the value of the study, and the claims that can be made based on its findings.  As noted above, the methodology and analysis is high quality, but seems to be almost too much for the research, seeming to over-analysing the small data set. 

For example, the list of hypotheses (P 6) is v long, and both not well explained in terms of the model you are seeking to use (how did you derive these hypotheses?) and I think too many in number to be supported by the analysis of such a small and specific data set.

Perhaps a shorter paper or research note, framed as initial findings, and on a specific location or form of ecotourism destination (wetlands) could be more appropriate? Or a further research project including more subjects and from a wider source to justify a full- journal article? 

Another approach would be to focus on a key finding - noted above and in your title - about local residents becoming local tourists, rather that the greater claims made of the findings in the current manuscript.

P 2 Line 5: '[w]hile it has developed rich experience in North America and Europe ecotourism is developing countries has been underdeveloped': the literature would contradict this contention with many studies of nature-based and ecotourism development in under- and less-developed countries. A useful source (including generally for the manuscript) would be Khanra et al (2021) 'Bibliomentric analysis and lit review of ecotourism: Toward sustainable dev.'

Minor and typographical issues

Title

I wonder if your proposed title is too long - both in terms of the substance of the paper (see above), but also in a practical sense;  perhaps something shorter like 'Local residents becoming local tourists: Value creation in Chinese wetland parks during the Covid-19 pandemic'

Language suggestions

P 3 line 5: perhaps change 'neglection' to something like 'focus'

P 11 line 23: perhaps re-phrase 'According to Table 4...' to something like 'Based on our regression analysis (as set in Table 4)...'.

P 13 line 19 - maybe replace 'six' with '6', and 'per annual' with 'annually'.

References

Capitalise journal titles in Refs 12, 28, 29, 36, 41, 50, 57, 65, 66, 73, 74, 83 & 84

Add journal title in ref 40

Add editors names in ref 13

Author Response

Dear professor,

Please accept our sincere thanks for your constructive comments and suggestions for our manuscript. We revised the manuscript according to your comments, specifically:

We followed your comment

Ref 2 is your source for detail on 'ecotourism': I think a more mainstream article with a stronger tourism focus, and including a suitable definition, such as Cobbinah (2015) 'Contextualising the meaning of ecotourism' would be more appropriate

And updated the definition in new Reference 2 (page 2).

Ref 4 relates to ecosystem services - maybe a more applicable source on this point would be say Keenan et al (2019) 'Ecosystem services in environmental policy', or Maes et al (2019) 'Ecosystem services are inclusive and deliver multiple values'.

We updated the definition in new Reference 3 (page 2).

P 5 Line 21 'We define tourist expectations as the results of ...': no source is provided for this statement, and I suggest this may be a contentious point which readers would need to be convinced of

We rephrased this as ‘Based on [38, 39], value co-creation intention can be understood as a local tourist’s intention to physically and virtually participate in ecotourism activities related to a specific wetland park destination.’ (page 6)

Ref 67, used at the foot of P 5 in regard to local tourists attitudes: this ref is written in the context of climate change and tourism.  While relevant I would seek down a more general source defining tourists attitudes.

We updated this into Ref 49 and Ref 50.

Methodology

For example, the list of hypotheses (P 6) is v long, and both not well explained in terms of the model you are seeking to use (how did you derive these hypotheses?) and I think too many in number to be supported by the analysis of such a small and specific data set.

We integrated hypotheses into the literature review (pp. 7-12)

Another approach would be to focus on a key finding - noted above and in your title - about local residents becoming local tourists, rather that the greater claims made of the findings in the current manuscript.

Thank you for the suggestion. We followed your suggestion to remove the overgenerlization of our findings and discussion, with the focus primarily on local residents as local tourists.

P 2 Line 5: '[w]hile it has developed rich experience in North America and Europe ecotourism is developing countries has been underdeveloped': the literature would contradict this contention with many studies of nature-based and ecotourism development in under- and less-developed countries. A useful source (including generally for the manuscript) would be Khanra et al (2021) 'Bibliomentric analysis and lit review of ecotourism: Toward sustainable dev.'

We updated the reference as ‘While it has developed rich experience in North America and Europe, ecotourism practices in developing countries have been underdeveloped [7].’ (page 2)

Minor and typographical issues

Title

I wonder if your proposed title is too long - both in terms of the substance of the paper (see above), but also in a practical sense;  perhaps something shorter like 'Local residents becoming local tourists: Value creation in Chinese wetland parks during the Covid-19 pandemic'

We have modified the title as ‘Local Residents Becoming Local Tourists: Value Co-creation in Chinese Wetland Parks During the COVID-19 Pandemic’

Language suggestions

P 3 line 5: perhaps change 'neglection' to something like 'focus'

We changed ‘neglection’ into ‘focus’ in page 4.

P 11 line 23: perhaps re-phrase 'According to Table 4...' to something like 'Based on our regression analysis (as set in Table 4)...'.

Changed into ‘Based on our regression analysis (as set in Table 4)’ on Page 20

P 13 line 19 - maybe replace 'six' with '6', and 'per annual' with 'annually'.

This part was removed from the manuscript to avoid overgeneralization.

References

Capitalise journal titles in Refs 12, 28, 29, 36, 41, 50, 57, 65, 66, 73, 74, 83 & 84

Modified

Add journal title in ref 40

Modified

Add editors names in ref 13

Modified

Kind regards,

Reviewer 4 Report

The article is interesting, correctly explained, and has a good review of previous research at local and international levels. 

Author Response

Dear professor,

Please accept our sincere thanks for your constructive comments and suggestions for our manuscript. We revised the manuscript according to your comments, specifically:

Following your comment ‘Put the contribution of the paper in the abstract’, we added on page 1:

‘Drawing on our survey in a specific wetland park, we highlight how local tourists’ attitude and perceived value positively affect their value co-creation intention and identify one more possible source of destination awareness: friends’ sharing of destination information and experience through social media. Practically, we suggest local tourism to offset maintenance costs of wetland parks during the COVID-19. That requires leveraging social norms and understanding residents’ expectations, in addition to improving infrastructure.’

In the introduction, emphasize the aim of the paper.

Given the above research gap, this study aims to draw on the theory of planned behavior (TPB) to examine the antecedents of local tourists’ value co-creation intentions during the COVID-19 pandemic in China, where international travel is still not available. (page 5)

Web sources in the paper go in the references and not in the footnotes.

They have been converted into reference 5, 6, 15, 16, 22

In Table 3, think about whether to calculate the correlation for the general characteristics of the respondents, namely: . Gender, Education, Age, Income and Location. So far I have not seen that put in this analysis. It is necessary to separate the discussion from the conclusion.

We have included them in Table 3 (page 19) now.

Kind regards,

Reviewer 5 Report

Greetings,

The paper is well written. All selections are good, methodology, results and other. Minor corrections should be made to make the paper suitable for publication. Put the contribution of the paper in the abstract. In the introduction, emphasize the aim of the paper. Web sources in the paper go in the references and not in the footnotes. In Table 3, think about whether to calculate the correlation for the general characteristics of the respondents, namely: . Gender, Education, Age, Income and Location. So far I have not seen that put in this analysis. It is necessary to separate the discussion from the conclusion.

All the best.

Author Response

(The authors gave the same response as above.)

Round 2

Reviewer 1 Report

Dear Authors,

Thank you for the careful review and additions done. Hope to read more work on this research topic as it is important for the coherent evaluation of conservation work done in protected ares and its proper funding and management.

Sincerely,

TI

Reviewer 3 Report

Thank you for your prompt response to comments and I now believe this is ready for publication.